# The Gambling Habits of University Students in Aragon, Spain: A Cross-Sectional Study

**DOI:** 10.3390/ijerph19084553

**Published:** 2022-04-09

**Authors:** Yolanda López-del-Hoyo, Alicia Monreal-Bartolomé, Pablo Aisa, Adrián Pérez-Aranda, Carlos Plana, José Antonio Poblador, Jaime Casterad, Javier García-Campayo, Jesus Montero-Marin

**Affiliations:** 1Department of Psychology and Sociology, University of Zaragoza, 50009 Zaragoza, Spain; ylopez.iacs@gmail.com (Y.L.-d.-H.); jgarcamp@gmail.com (J.G.-C.); pabo.aisa@gmail.com (P.A.); 2Aragon Institute for Health Research, IIS Aragon, 50009 Zaragoza, Spain; aliciamonbart@usal.es; 3Primary Care Prevention and Health Promotion Research Network, RedIAPP, 28029 Madrid, Spain; 4Department of Basic, Developmental and Educational Psychology, Autonomous University of Barcelona, 08193 Cerdanyola del Vallès, Spain; 5Department of Physical Medicine and Nursing, University of Zaragoza, 50009 Zaragoza, Spain; carplana@unizar.es (C.P.); pobla@unizar.es (J.A.P.); jcaster@unizar.es (J.C.); 6Department of Psychiatry, University of Zaragoza, 50009 Zaragoza, Spain; 7Department of Psychiatry, Warneford Hospital, University of Oxford, Oxford OX3 7JX, UK; jesus.monteromarin@psych.ox.ac.uk; 8AGORA Research Group, Teaching, Research & Innovation Unit, Parc Sanitari Sant Joan de Déu, 08830 St. Boi de Llobregat, Spain

**Keywords:** gambling, pathological gambling, addictive behavior, university students

## Abstract

Gambling has become a routine form of entertainment for many young people. The aim of this study was to describe the gambling behavior that university students are developing in Aragon, Spain, and to analyze whether these habits are more common among students of sports science, on the assumption that they are more likely to have a higher exposure to betting company marketing. A cross-sectional design was applied, with data collected on advertising exposure, gambling habits and experiences, and opinions on the impact of gambling and its regulation from 516 undergraduate students from the University of Zaragoza. The online survey included ad hoc questions and the “Pathological Gambling Short Questionnaire” to screen for potential gambling disorders. Almost half of the sample had bet money at least once in their life (48.1%), and 2.4% screened positive for consideration of a possible diagnosis of pathological gambling. Betting shops (44.2%) were the most common gambling option, and students of sports science showed a higher prevalence of pathological gambling and had greater tendencies to make bets. Gambling is perceived as a normal leisure activity by a significant part of university students. The development of transversal strategies is required to raise awareness towards the potential dangers of gambling.

## 1. Introduction

Gambling has increasingly become a normalized activity. According to a Spanish government report of 2020, 64.2% of the Spanish population admitted to having gambled at least once in the previous 12 months [1]. These percentages have increased over the years, which can be partly explained by the direct effect of advertising: the expenditure on gambling advertising in Spain amounted to €170 million in 2018, with a 150% increase in the previous five years [2]. The main aim of betting marketing is to make gambling an acceptable leisure activity and even a socially desirable behavior [3,4]. Advertising campaigns typically depict gambling as an interesting and enjoyable activity [5] in which “easy money” can be made [6]. This makes it extremely appealing for young adults with very limited or no income, such as many university students, who can see gambling as a financial opportunity, added to other possible factors such as seeking entertainment, socializing, or mitigating negative emotional states [7].

Increasing exposure to advertising can represent a serious risk factor for the development of pathological gambling, as many studies suggest an association between exposure to gambling advertising and the increase of gambling frequency in young people [8,9,10]. In turn, high-frequency gambling is an important risk factor for the development of pathological gambling [10,11]. In August 2021, a new Spanish law (Royal Decree 958/2020) included new regulations to establish the conditions permitted for the advertising, sponsorship, and promotion of gambling activities. These regulations should reduce the exposure of the most vulnerable individuals to betting marketing, but such messages can still be received under certain circumstances (e.g., registered and verified users of gambling sites) and, consequently, there will still be a certain degree of exposure that can imply a risk factor for developing addictive behaviors.

According to epidemiological studies, young people (aged under 30) represent one of the most common age groups to suffer from pathological gambling [12]. Moreover, most patients had their first gambling experiences between the ages of 19 and 23 years of age [13], which implies that these ages represent key moments in the individual’s relationship with gambling. As previously stated, these people, who are often studying at university or just starting out in their careers, are at high risk of finding gambling appealing, mainly because of the possibility of considering it an economically profitable activity. In turn, this belief is strongly associated with the development of pathological gambling [14,15].

In addition to the traditional gambling options (e.g., betting shops, casinos, bingo halls), other types of gambling, mainly delivered via Internet sites, have become increasingly popular in the last decade, mainly among young people. This is the case of micro-transactions, in-game purchases commonly included within online videogames that consist of small payments, sometimes in the form of a bet, to obtain a virtual item. Another type of gambling that has proliferated in recent years is sports betting [1]. McGee [16] has identified some key elements for the normalization of this behavior, which is more common among men, such as the anonymity and the incentives and in-play promotions that sports gambling platforms offer via mobile applications. Furthermore, exposure to sports betting marketing is often the norm on different platforms, mainly during live and televised sporting events, where the logos of betting establishments are displayed on players’ uniforms, team banners, scoreboards, and advertising hoardings around playing fields [17], although this is no longer allowed under the new Spanish legislation. The so-called “gamblification” of sport [10], which takes advantage of fan support and sports team loyalty, enhances gambling normalization in young people through a context of family-friendly leisure [18,19]. This implies that individuals who enjoy watching sports might have been more exposed to betting marketing and, therefore, could be at a higher risk position for developing gambling disorders.

In this context, the aim of the present study was to explore and describe the gambling behaviors of a large sample of university students in the region of Aragon, Spain, including the level of exposure to betting marketing, the frequency with which they use different gambling options (i.e., microtransactions, sports betting, betting shops, and betting websites), and their opinions on the effects of gambling and how it should be regulated. Moreover, considering the potential higher risk of exposure for those individuals enjoy watching sporting events, previously described, the gambling habits of a subsample of undergraduate students of sports science was compared to those of the other university students. The hypothesis behind this was that this subsample would have been more exposed to betting marketing and, subsequently, would present with a higher frequency of different gambling habits, including sports betting.

## 2. Materials and Methods

### 2.1. Participants

Undergraduate students from the University of Zaragoza enrolled in 25 different degree courses were invited to participate in this cross-sectional study. In accordance with the guidance offered by the Transparency Portal of the University of Zaragoza (2019), the sample recruitment process aimed to achieve the highest degree of similarity to the general population in terms of sex, city, and area of knowledge. The inclusion criteria for participating were being a student enrolled at the University of Zaragoza, residing in one of the provinces comprising the Aragon region, and signing the informed consent form. No international students participated in the present study.

### 2.2. Instruments

The complete survey is included in the Appendix A. After answering questions on their age, gender, city, and course, the participants were presented with questions regarding their exposure to betting advertising and gambling prevention messages. They were then asked questions about their knowledge and use of microtransactions, sports betting, betting shops, and betting websites. For each case, the participants were asked how often they used that service, how much money they would spend, and how they felt while gambling, among other questions. Next, a number of questions were asked regarding the participant’s opinions on the effects of gambling and its regulation.

Finally, the participants answered the *Cuestionario Breve de Juego Patológico* (“Pathological Gambling Short Questionnaire”, or CBJP by its Spanish initials) [20], which consists of four dichotomous items (“yes” or “no”). The cutoff for screening positive for pathological gambling is set at 2 points, with 100% sensitivity and 97.5% specificity.

In addition, four items that were considered particularly significant for the study were included; three of these referred to common risk factors for developing pathological gambling and were extracted from or adapted from validated measures: “Have you gambled more than you intended to?” (South Oaks Gambling Screen, SOGS) [21]; “Have you ever neglected your obligations (e.g., family, work or school) because of gambling?” (Massachusetts Gambling Screen, MAGS) [22]; and “Have you lost track of time while gambling?” (Online Pathological Gambling Scale, O-PGS) [23]. Finally, the item “Have you been told (or have you thought) of asking for professional help?” (MAGS) was included to directly assess the self-awareness level of individuals who could present with pathological gambling.

### 2.3. Procedure

The study was conducted during the 2018–2019 academic year, and the data were collected via Google Forms between April and May 2019. The online survey was completed digitally in front of the researcher in charge of the project, who asked faculty members for permission to interrupt their classes for a few minutes. In some cases, the faculty members themselves were entrusted with presenting the survey. No personal information was asked of participants when completing the survey, and all of the collected data were used only for the purpose of completing this research project. This study was approved by the Research Ethics Committee of Aragon (reference number: PI19/280).

### 2.4. Data Analysis

The cross-sectional nature of this study implied the performance of descriptive analyses, reporting means (M) and standard deviations (SD) for continuous variables, and frequencies and percentages for categorical variables. In addition, the Chi-square test and Student’s *t*-test were performed to analyze potential differences in gambling habits between students enrolled in sports science courses and the rest. Multiple linear and logistic regressions were calculated to explore the predictive role of the study of sports science along with sociodemographic variables on the number of risk factors for pathological gambling, and on screening positive in the CBJP. An alpha level of 0.05 was set, using a two-tailed test. Data analyses were computed using SPSS v26.0 (SPSS Inc., Chicago, IL, USA).

## 3. Results

### 3.1. Sociodemographic Characteristics of the Sample

The total sample included 516 undergraduates with a mean age of 20.57 years (SD = 2.37) and a higher proportion of females (*n* = 314, 61%). A considerable proportion of participants were studying health-related courses (*n* = 159, 30.8%). Most of the sample lived in an urban setting (*n* = 415, 81.5%), and the average income of their district of residence was mostly between €10,000 and €15,000 per year (*n* = 225, 64.8%). Most students reported having less than €100 (*n* = 240, 46.5%) to spend on their monthly expenses. With regard to their lifestyles and interests, 332 students (64.3%) reported engaging in sport at least every week, and approximately one third of the sample expressed considerable interest in sport (*n* = 171, 33.1%). Table 1 summarizes the sociodemographic characteristics of the total sample and the two subgroups for comparison.

### 3.2. Exposure to Gambling Advertising and Prevention Messages

Gambling advertisements was received daily by 60 students (11.6%), although most participants reported “sometimes” receiving this type of advertising. On the other hand, approximately half of the sample (*n* = 279, 54.1%) said they had never received any prevention messages regarding the dangers of gambling. The family, the internet, and high school were the most common sources of prevention messages (see Table 2).

### 3.3. Betting and Gambling Experiences

Almost half of the students admitted to having bet money (regardless of the type of gambling option) at least once in their life (*n* = 248, 48.1%). The main reason for gambling was entertainment (*n* = 112, 45.2%), followed by the possibility of winning money (*n* = 88, 35.5%). Considering their answers to the CBJP, 12 (2.4%) presented scores for a possible diagnosis of pathological gambling, of whom 10 acknowledged having a gambling problem and one expressing the belief that he needed professional help; 8.1% of the sample (*n* = 42) reported having gambled more money than they intended; 2.7% (*n* = 14) admitted to having neglected their obligations because of gambling; and 7.2% (*n* = 37) had lost track of time while gambling.

The use of microtransactions, sports betting (including live betting), betting shops, and betting websites was analyzed separately, and some of this information is summarized in Table 3. While 216 students (41.9%) did not use any of these gambling options, 108 (20.9%) used one, 97 (18.8%) used two, 63 (12.2%) used three, and 32 students (6.2%) used all four gambling options:

The use of microtransactions was relatively frequent in our sample, as 133 participants (25.8%) admitted to having paid small amounts to play online games on their smartphones, computers, or tablets. On certain occasions (*n* = 39, 29.3%), the money spent on these micro-transactions ended up exceeding €30.

A larger proportion of the sample had taken part in sports betting (*n* = 200, 38.8%). While almost 25% of these participants had received a welcome bonus (*n* = 48), which varied greatly in amount (€5–€200), this incentive was given relatively little importance (M = 4.00, SD = 3.38). For those who bet on different sports, the average weekly expenditure was €3.18 (SD = 15.83), while the total expenditure was €123.72 (SD = 303.45) of the different forms of sports betting, the specific case of live betting was used by 116 students (22.5%). Soccer was the most common sport, followed by basketball, and tennis. A considerable proportion of the sample (*n* = 151, 29.3%) admitted using live bets on sports on specific occasions, while 42 (8.2%) reported a frequency of at least once per month. Among those who used the live betting option, 43 participants (37%) admitted that they used benefits or promotions provided by the app to continue betting.

With regard to betting shops, a large number of students (*n* = 356, 69%) acknowledged having entered one of these establishments at least once, whereas close to half of the sample (*n* = 228, 44.2%) reported having placed a bet at a betting shop or through a bookmaker. The mean age of their first experience was 18.31 years (SD = 1.72), although 52 participants (32.5%) were under 18 when they entered a betting shop for the first time. The main reason for entering a betting shop for the first time was to accompany a friend (*n* = 211, 40.9%), while a significant proportion admitted doing so out of curiosity (*n* = 98, 19%). Currently, 77 students (14.9%) admitted entering betting shops on specific occasions, and 40 (7.6%) said that they did so on a monthly or weekly basis. Only two participants (0.4%) admitted doing so daily.

Inside betting shops, sports were very frequent gambling options (*n* = 156, 68.4%); however, roulette was the most popular (*n* = 188, 82.5%), while bingo games (*n* = 61, 26.8%) and slot machines (*n* = 35, 15.4%) were not as common. The average weekly expenditure at these establishments was €5 (SD = 20.82). A considerable proportion of those who gambled (*n* = 91, 39.9%) reported having won substantial amounts on occasion. As far as emotional states are concerned, some of the students stated that they had felt indifferent before gambling or betting (*n* = 71, 31.1%), while some others admitted to feeling entertained (*n* = 69, 30.3%), nervous (*n* = 44, 19.3%), and excited (*n* = 27, 11.8%). These emotions subsequently changed during the game, when most of the participants referred feeling nervous (*n* = 115, 50.4%), and at the end of the experience, some of them felt sad or disappointed (*n* = 61, 26.8%), while some others went back to feeling indifferent (*n* = 68, 29.8%). With regard to the routines of the individuals who gambled, afternoon and night were the preferred times of the day for 168 (73.7%) and 109 (47.8%) participants, respectively. Finally, taking the whole sample into consideration, 41 people (7.9%) admitted to thinking about gambling or betting “sometimes, mostly on the weekends”, and only 14 (2.7%) reported thinking about gambling on a daily basis.

With regard to betting websites, 103 individuals (20%) reported having entered one such websites at least once, and 58 students (11.2%) had bet some money through one of these websites at least once. The average age for doing so for the first time was 18.16 years (SD = 1.89), although a small proportion of the sample (8.8%) were under 18 at the time. Most of the sample did not have an account on any betting website (*n* = 462, 89.5%). In most cases, individuals who had used betting websites had given their real identification details (*n* = 54, 52.4%), and most of them had received a welcome bonus (*n* = 42, 72.4%), which was generally a small amount (€5–€20) and was not given much importance (M = 3.90, SD = 3.54). Among those who used betting websites, only four individuals (6.9%) admitted to betting several times per week.

On betting websites, betting on different sports was the most popular option (*n* = 54, 93.1%), followed by roulette (*n* = 24, 41.4%) and poker (*n* = 11, 19%) games. The weekly expenditure on these websites by those who gambled was relatively low on average (M = €6.45, SD = €29.23), and almost half of them reported having won substantial amounts at least once (*n* = 25, 43.1%). With regard to the emotional states experienced by individuals who gambled on websites, a significant part reported feeling nervous before placing the bet (*n* = 18, 31%), but this feeling was more common during the online gambling experience (*n* = 27, 46.6%). Afterwards, most described themselves feeling “satisfied” (*n* = 16, 27.6%) or “indifferent” (*n* = 18, 31%). As far as the routines of the individuals using these websites is concerned, afternoon (*n* = 48, 82.8%) and night (*n* = 35, 60.3%) were the preferred times for betting. Finally, considering the whole sample, only a small proportion admitted to thinking about online betting with any frequency. Six students (1.2%) said that they did it “sometimes, mostly on the weekends”, and just two (0.4%) admitted to thinking about it daily.

### 3.4. Knowledge and Opinions on Gambling

Only 31 students (6%) were not able to locate any betting shops in their district, while 122 (23.6%) said that they could locate many of them. Likewise, a small proportion (*n* = 71, 13.8%) reported not knowing any betting websites, while 247 (47.9%) knew a few. Most of the sample (*n* = 386, 74.8%) reported that they knew at least one friend who gambled regularly. A significant proportion said that these friends had won “large or relatively large amounts of money” (*n* = 305, 59.1%).

With regard to the students’ opinions on betting, the vast majority (*n* = 472, 91.5%) agreed with the sentence “young people are betting on sports frequently”, while most of them also shared this belief regarding young people’s use of betting shops (*n* = 433, 83.9%) and betting websites (*n* = 388, 75.2%). Moreover, most of the participants did not agree with the statement “there are few physical locations where bets can be placed” (*n* = 474, 91.9%), but they were more divided in their opinion on whether betting was a “normal activity” (yes: *n* = 232, 45%). A considerable proportion of the sample believed that betting advertising should be banned (*n* = 199, 38.6%), while approximately half of the participants thought that stronger restrictions should be applied (*n* = 277, 53.7%). On average, gambling was perceived by our sample as a dangerous activity (M = 8.08, SD = 1.75).

### 3.5. Students of Sport Science vs. the Others

Significant differences were observed in sociodemographic variables when comparing sports science students with the others (see Table 1): age (t = −2.77, *p* = 0.006), sex (χ^2^ = 21.92, *p* < 0.001), year (χ^2^ = 22.97, *p* < 0.001), residence (χ^2^ = 12.51, *p* < 0.001), money for expenses (χ^2^ = 33.57, *p* < 0.001), interest in sport (χ^2^ = 146, *p* < 0.001), and weekly engaging in sport (χ^2^ = 55, *p* < 0.001).

The CBJP total score did not present significant differences compared to the rest of the sample (t = −1.50, *p* = 0.136), and this remained statistically insignificant after including the sociodemographic variables that presented significant differences between the two subgroups as covariates. However, there was a significantly higher proportion of individuals who screened positive among sport science students: 5.1% vs. 1.7% (χ^2^ = 3.89, *p* = 0.049). Another significant difference was also observed in one of the additional items for assessing risk factors: A higher proportion of individuals who had neglected their responsibilities was observed among those who studied sport science (6% vs. 1.9%, χ^2^ = 4.96, *p* = 0.026).

With regard to degree of exposure to gambling advertisements, no significant differences were observed (χ^2^ = 3.48, *p* = 0.323). Nonetheless, in terms of gambling habits, sports science students used, on average, more gambling options than the rest (t = 4.56, *p* < 0.001), even after including the covariates in the model (F = 5.65, *p* = 0.018). Moreover, a significantly higher proportion of individuals who used microtransactions was found among this subsample: 36.6% vs. 23.1% (χ^2^ = 7.74, *p* = 0.005), although they did not spend significantly larger amounts of money. Also, a significantly higher proportion of those who studied sport science made use of sports betting (60.6% vs. 33.5%; χ^2^ = 24.77, *p* < 0.001) and live betting (41.6% vs. 17.9%; χ^2^ = 26.16, *p* < 0.001). In addition, they spent significantly larger amounts of money on sports betting (t = −2.69, *p* = 0.009). Along the same lines, this subsample contained a larger proportion of individuals who had entered betting shops (78.2% vs. 66.7%; χ^2^ = 4.99, *p* = 0.025) and betting websites (32.7% vs. 16.9%; χ^2^ = 12.70, *p* < 0.001). No significant differences were found between the subsamples in terms of knowledge of betting websites and locations of betting shops, gambling promotion or prevention messages, or in their opinions regarding the impact of gambling and its regulation.

The regression models showed that studying sport science was not a significant predictor of the number of risk factors presented for pathological gambling (CBJP + 4 additional items) nor of screening positive in the CBJP. The only sociodemographic variable that consistently predicted both outcomes was sex (B = 0.64, *p* < 0.001 for risk factors and B = 2.90, *p* = 0.006 for the screening).

## 4. Discussion

The main aim of the present study was to describe the gambling habits of university students in Aragon, Spain. According to our results, betting shops (44.2%) were the most common option used for gambling, much more than online betting (11.2%). This is in agreement with the 2020 report on behavioral addictions published by the Spanish Ministry of Health [1], which observed that in-person gambling options such as betting shops had been much more commonly used (63.6%) than online betting platforms (6.7%) in the previous 12 months. While those results are not fully comparable to the ones of the present study due to the different age groups (i.e., university students vs. people aged 15–64) and the time frame (i.e., lifetime vs. previous 12 months) analyzed, they seem to suggest that online betting is still not as popular as the traditional in-person gambling options, and therefore that preventive strategies should not overlook these targets despite the increasing significance of online platforms in behavioral addictions.

Our results suggest that the degree of normalization of gambling among this sample was relatively high, not only because of the relatively high frequency of students who had gambled (almost 50%) but also due to their opinions––45% of the students literally considered gambling as a “normal activity”––and their knowledge on different gambling options (e.g., betting shops in their districts, websites, etc.). This conforms to the effect sought by betting advertising, which endeavors to present gambling as a normal and fun leisure activity [3,4,5]. In our sample, most students were exposed at least “sometimes” to gambling advertisements, which have been shown to have a direct impact on gambling behaviors in young adults [8,9,10].

Nonetheless, as previously pointed out by Parrado-González and León-Jariego [10], these advertisements seem to have more effect on the normalization of gambling than on favorable attitudes, as most of the students presented negative opinions towards gambling: it was perceived as dangerous, and most students considered that gambling advertising should be regulated or even banned. The recently enacted Spanish law regulating the advertisement, sponsorship, and promotion of gambling activities advocates making legal gambling less visible to young adults, in agreement with what has been considered the best strategy to cope with this emerging problem [24,25,26]. In this regard, establishing strict controls on the legal age for gambling has been identified as an effective measure [26]. Considering the answers given by our participants, fewer than 10% of the sample had engaged in formal gambling (e.g., betting shops, betting websites) before turning 18, the legal age in Spain, which represents a lower proportion than the percentages reported in other studies [27,28,29].

Exposure to gambling advertisements can contribute to the development of pathological gambling [10,11], which presented a relatively high prevalence in our sample (2.4%). Of the reasons for gambling given in our study, the possibility of winning money was relatively common (35.5% of participants who gambled). Again, this shows that gambling advertising is very likely to have a direct effect on the popularity of this belief, which is a risk factor for developing pathological gambling [14,15]. It is also noteworthy that many students reported knowing someone who gambled regularly, and that these acquaintances had allegedly won significant amounts of money through this activity. This belief is probably a misconception produced by cognitive biases that enhance positive inputs while ignoring negative experiences related to gambling. Thus, we agree with León-Jariego et al. on the importance of designing educational programs that teach young adults about the low impact self-perceived gambling skills have on winnings from gambling and debunk misconceptions about its profitability [30].

Finally, the present study hypothesized that students enrolled in sport science courses would be more exposed to betting marketing related to sporting events, which would increase their interest in gambling. Contrary to expectations, this subsample was not significantly more exposed to betting advertisements, which refutes the first part of our hypothesis. Nonetheless, sport science students did resort to different gambling options more often than other students did, and they spent significantly larger amounts of money on bets. Moreover, a significantly higher prevalence of pathological gambling was observed among these students (5.1%). Although exposure to betting advertising was not higher for this subsample, it is possible that sport science students were more exposed to other types of publicity (logos on players’ uniforms, team banners, scoreboards, etc.) which could also have an impact on their interest in gambling [17]. In addition, these students may feel like their expertise in sport makes them more likely to succeed, and “being good at gambling” may become an achievement in itself [31,32]. However, it should be noted that the sociodemographic profile of the sport science students in this study presented some aspects that have been commonly associated with regular gambling, such as a larger representation of rural settings [33] and a higher proportion of men [13,16]. In this regard, while studying sport science and other sociodemographic variables were not significant predictors of the number of risk factors nor of screening positive for pathological gambling, sex did play a significant role in both cases, confirming the higher prevalence of this problem among men, even in the university student population [34], which could be the reason behind the differences between the two subgroups that we compared.

Certain limitations of the present study need to be acknowledged. First, the study used a convenience sample of university students in Aragon, Spain, and therefore the generalizability of the results is limited. Moreover, data were collected through self-report measures, which affects the ecological validity of the study, and many ad hoc questions were included, although they were all based on a comprehensive review of the literature after considering validated measures (e.g., MAGS and SOGS). In this respect, it should also be noted that the measure used for assessing potential pathological gambling in this study, CBJP, is a screening measure, and that a proper diagnosis according to DSM-5 criteria should have been included to increase the validity of the outcome. A number of other items could have been included to assess risk factors, but the length of the survey obliged the researchers to reduce this section. Likewise, other important variables that could have probably added value to our descriptive exploration were not assessed, such as impulsivity, mood, anxiety, family support, and substance/alcohol use disorders, which have commonly been related to gambling behaviors [35,36,37]. Future studies could include those variables and analyze how they can predict the frequency of use of different gambling options, money spent, and the risk of developing pathological gambling. Finally, should also be noted that this study was conducted before the implementation of the new Spanish law governing betting advertising (August 2021), and therefore it should be replicated by future research studies to assess the potential impact of the new law.

## 5. Conclusions

This study has corroborated that gambling is perceived as a normal leisure activity by a significant proportion of university students, and that microtransactions, sports betting, betting shops, and websites are services that these individuals use with relative frequency. The prevalence of pathological gambling was set at 2.4%, which highlights that this population is in high risk of developing the disorder, as previous research has already indicated. In addition, sport science students present an even higher prevalence of pathological gambling and significantly higher frequencies of use of the different gambling options, which suggests that this population is probably affected by different risk factors that makes it particularly vulnerable. Therefore, in addition to the proper regulation of betting advertising, there is a need for the development of transversal strategies that are not restricted to teenagers and young adults, but also include families and educational institutions, such as universities, in order to raise awareness of the potential dangers of gambling.

## Figures and Tables

**Table 1 ijerph-19-04553-t001:** Sociodemographic characteristics of the sample.

Variable	Total Sample (*n* = 516)	Sports Science Students (*n* = 101)	Other Students (*n* = 415)
**Age**, M (SD) [*n* = 515] *	20.57 (2.37)	21.15 (2.34)	20.42 (2.36)
**Sex** [*n* = 515] *			
Male, *n* (%)	201 (39%)	60 (59.4%)	141 (34.1%)
Female, *n* (%)	314 (61%)	41 (40.6%)	273 (65.9%)
**Field of study** [*n* = 516]			
Health sciences, *n* (%)	159 (30.8%)		
Social and juridical sciences, *n* (%)	129 (25%)		
Sports science, *n* (%)	101 (19.6%)		
Engineering/architecture, *n* (%)	65 (12.6%)		
Arts and humanities, *n* (%)	31 (6%)		
Other sciences, *n* (%)	31 (6%)		
**Year** [*n* = 509] *			
1st, *n* (%)	149 (29.2%)	33 (32.7%)	116 (28.3%)
2nd, *n* (%)	131 (25.6%)	22 (21.8%)	109 (26.6%)
3rd, *n* (%)	183 (35.8%)	25 (24.8%)	158 (38.5%)
4th, *n* (%)	48 (9.4%)	21 (20.8%)	27 (6.6%)
**District of residence** **average income** [*n* = 347]			
<€10,000, *n* (%)	48 (13.8%)	6 (19.4%)	42 (13.3%)
€10,000–€15,000, *n* (%)	225 (64.8%)	22 (71%)	203 (64.2%)
>€15,000, *n* (%)	74 (21.3%)	3 (9.7%)	71 (22.5%)
**Type of area of residence** [*n* = 509] *			
Rural, *n* (%)	94 (18.5%)	31 (30.7%)	63 (15.4%)
Urban, *n* (%)	415 (81.5%)	70 (69.3%)	345 (84.6%)
**Money for expenses** [*n* = 515] *			
<€100, *n* (%)	240 (46.5%)	25 (24.8%)	215 (51.9%)
€101–€400, *n* (%)	215 (41.7%)	58 (57.4%)	157 (37.9%)
€401–€600, *n* (%)	39 (7.6%)	16 (15.8%)	23 (5.6%)
€601–€1000, *n* (%)	14 (2.7%)	2 (2%)	12 (2.9%)
>€1001, *n* (%)	7 (1.4%)	0 (0%)	7 (1.7%)
**Interest in sport** [*n* = 516] *			
None, *n* (%)	55 (10.7%)	1 (1%)	54 (13%)
A little, *n* (%)	172 (33.3%)	0 (0%)	172 (41.1%)
Some, *n* (%)	118 (22.9%)	17 (16.8%)	101 (24.3%)
Considerable, *n* (%)	171 (33.1%)	83 (82.2%)	88 (21.2%)
**Engaging in sport/week** [*n* = 516] *			
No, *n* (%)	184 (35.7%)	4 (4%)	180 (43.4%)
Yes, *n* (%)	332 (64.3%)	97 (96%)	235 (56.6%)

* Means statistically significant *p*-values.

**Table 2 ijerph-19-04553-t002:** Summary of sample’s exposure to advertising and prevention messages.

	Total Sample (*n* = 516)	Sports Science Students (*n* = 101)	Other Students (*n* = 415)
**Frequency of exposure to betting advertisements**			
Never, *n* (%)	46 (8.9%)	8 (7.9%)	38 (9.2%)
Very rarely, *n* (%)	127 (24.6%)	30 (29.7%)	97 (23.4%)
Sometimes, *n* (%)	141 (27.3%)	28 (27.7%)	113 (27.2%)
Often, *n* (%)	99 (19.2%)	21 (20.8%)	78 (18.8%)
Very often, *n* (%)	42 (8.1%)	6 (5.9%)	36 (8.7%)
Every day, *n* (%)	60 (11.6%)	8 (7.9%)	52 (12.5%)
**Exposure to preventive information**			
Yes, *n* (%)	279 (54.1%)	62 (61.4%)	217 (52.3%)
No, *n* (%)	236 (45.7%)	39 (38.6%)	197 (47.5%)
**Preventive message source**			
Internet, *n* (%)	129 (46.2%)	34 (33.7%)	95 (22.9%)
Parents or family, *n* (%)	142 (50.9%)	27 (26.7%)	115 (27.7%)
School/university, *n* (%)	100 (35.8%)	20 (19.8%)	80 (19.3%)
Other, *n* (%)	53 (19%)	12 (11.9%)	41 (9.9%)

**Table 3 ijerph-19-04553-t003:** Summary of the use of different gambling options.

	Use, *n* (%)	Received Bonus, *n* (%)	Weekly Money Spent, M (*SD*)
Microtransactions	133 (25.8%)	-	-
Sports betting	200 (38.8%)	48 (24%)	€3.18 (15.83)
Betting shops	160 (44.2%)	-	€5 (20.82)
Betting websites	103 (11.2%)	42 (40.8%)	€6.45 (29.23)

## Data Availability

The data presented in this study are available on request from the corresponding author.

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
