# Peer review of "The Gambling Habits of University Students in Aragon, Spain: A Cross-Sectional Study"

_ijerph, 2022, doi:10.3390/ijerph19084553_

Round 1

Reviewer 1 Report

This manuscript covers an interesting topic in behavioral addictions, understanding gambling in students is a highly important issue as this population may benefit from specific approaches. The current study provides information about a specific scenario in a specific region. Although some of the results may be generalizable, for general readers (non-Spain readers), the main findings are related to the comparison among the two groups of students (sports sciences vs. others), as the other results have been studied in other countries with their own features and realities. In any event, there are some comments/suggestions/concerns that the authors must clarify in order to improve the manuscript’s quality:

MINOR CHANGES

- There are some important studies on the prevalence of gambling in Spain that are not mentioned. Specifically, the authors mention at the beginning of the introduction the results of a (outdated) study published in 2015, however, the “INFORME SOBRE Adicciones comportamentales 2020” is not mentioned nor discussed. This report provides the most recent information on gambling in Spain and the data may improve the introduction and discussion when this information is contrasted with the results.

(https://pnsd.sanidad.gob.es/profesionales/sistemasInformacion/sistemaInformacion/pdf/2020_Informe_adicciones_comportamentales.pdf)

MAJOR CHANGES

- The authors should clarify how the sample was selected to participate. To my understanding, the only inclusion criterion was to be a student of the University of Zaragoza, and the researchers collected the sample by interrupting the classes and inviting the subjects. Most of the main universities in Spain have international students (e.g. Erasmus students), were those students invited to participate? did merely local students participate? how did the researchers substantiate that only local students participated?. This point should be clarified as the authors try to analyze gambling in their location, but if there were students from other countries, cultures, and languages (even if they had proficiency in Spanish) may impact the current results.

On the other hand, the sample was collected among students that were face-to-face classes. What about online students? I understand that during the study period virtual classes were more uncommon than nowadays with COVID pandemic. However, students in virtual classes may have different profile than face-to-face students regarding gambling.

- May the authors provide some data about nicotine use, alcohol, or other substance use? Substance use disorders (SUD) and any mental disorder are frequently related to behavioral disturbances and behavioral addictions. If information about substance use is not available, it should be discussed and stated as a limitation.  

- The statistical analyses applied are fairly simple and restricted to mere group comparisons (only univariate and bivariate analyses were applied). The authors could execute multivariate methods (e.g. multiple regression analyses) that would allow for an adjusted testing of hypotheses and for estimating the sizes of the effects found.

- Could the authors discuss the possible reasons that the current sample has more betting online use than the prevalence of betting online in the general population in Spain in the last 12 months in 2019 (both, in the 15-65 years old group and 14-18 years old group; according to the "Observatorio Español de las Drogas y las Adicciones, 2020")

(https://pnsd.sanidad.gob.es/profesionales/sistemasInformacion/sistemaInformacion/pdf/2020_Informe_adicciones_comportamentales.pdf)

- The power to generalize the current results to the other populations may be affected as only university students were included (and was a convenience sample). Hence, the authors should state this as a limitation, as the current results may be only interpreted in specific similar samples.

Author Response

MINOR CHANGES

- There are some important studies on the prevalence of gambling in Spain that are not mentioned. Specifically, the authors mention at the beginning of the introduction the results of a (outdated) study published in 2015, however, the “INFORME SOBRE Adicciones comportamentales 2020” is not mentioned nor discussed. This report provides the most recent information on gambling in Spain and the data may improve the introduction and discussion when this information is contrasted with the results.

Authors: We completely agree with the reviewer that this 2020 update should have been included in our study. Actually, a new Spanish legislation on betting advertisements has been recently implemented (August 2021), and although this study was conducted before this new law (spring 2019), we have added some notes on this regard to update the information in the introduction and in the discussion.

The introduction includes data of the 2020 report (lines 39-40):

According to the Spanish government, in 2020, 64.2% of the Spanish population ad-mitted having gambled at least once in the previous 12 months [1].

The updated information regarding the Spanish legislation has been included (lines 54-60):

Since August 2021, the Spanish legislation (Royal decree 958/2020) included new regulations to establish the conditions that may be included in the advertisement, sponsorship, and promotion of gambling activities. These regulations should decrease the exposure of the most vulnerable individuals to betting marketing, but such messages can still be received under some circumstances (e.g., registered and verified users of gambling platforms) and, therefore, there will still be a certain degree of exposure that can imply a risk factor for developing addictive behaviors.

In the Discussion, we have noted that this study was conducted before the new regulations (lines 383-386):

Finally, it needs to be noted that this study was conducted before the implementation of the new Spanish legislation on betting advertisings (August 2021), and therefore it should be replicated by future research studies to assess the potential impact of the new regulations.

MAJOR CHANGES

- The authors should clarify how the sample was selected to participate. To my understanding, the only inclusion criterion was to be a student of the University of Zaragoza, and the researchers collected the sample by interrupting the classes and inviting the subjects. Most of the main universities in Spain have international students (e.g. Erasmus students), were those students invited to participate? did merely local students participate? how did the researchers substantiate that only local students participated?. This point should be clarified as the authors try to analyze gambling in their location, but if there were students from other countries, cultures, and languages (even if they had proficiency in Spanish) may impact the current results.

Authors: Thank you for pointing this out. The assessment process was exactly the one that the reviewer describes. International students (who represented a small percentage, around 2% in 2019, of the University of Zaragoza) did not participate in this study. This has been included in lines 103-105:

The inclusion criteria for participating were being a student from the University of Zaragoza, living in a province of Aragon, and signing the informed consent form. No international students participated in the present study.

On the other hand, the sample was collected among students that were face-to-face classes. What about online students? I understand that during the study period virtual classes were more uncommon than nowadays with COVID pandemic. However, students in virtual classes may have different profile than face-to-face students regarding gambling.

Authors: Since the data were collected during 2019, this was not the case. All the classes were conducted face-to-face.

- May the authors provide some data about nicotine use, alcohol, or other substance use? Substance use disorders (SUD) and any mental disorder are frequently related to behavioral disturbances and behavioral addictions. If information about substance use is not available, it should be discussed and stated as a limitation. 

Authors: We agree with the reviewer that these variables are relevant. They were not included in the present study since it was only focused on gambling behaviors, but we admit that this is a limitation of our research. This has been added in lines 378-381 along with a new reference (36):

In the same line, some relevant variables that could have probably added value to our descriptive exploration were not assessed, such as impulsivity, mood, anxiety, family support, and substance/alcohol use disorders, which have commonly been related to gambling behaviors [34–36].

Rash, C.; Weinstock, J.; Van Patten, R. A review of gambling disorder and substance use disorders. Subst. Abuse Rehabil. 2016, 7, 3.

- The statistical analyses applied are fairly simple and restricted to mere group comparisons (only univariate and bivariate analyses were applied). The authors could execute multivariate methods (e.g. multiple regression analyses) that would allow for an adjusted testing of hypotheses and for estimating the sizes of the effects found.

Authors: We agree with the reviewer, the analyses reported are descriptive and subgroup comparisons; that is because this study had the main aim of describing the behavior of the sample. Looking at the survey, it has many questions on the frequency of use of different types of gambling options, and questions related to each type (e.g., first experience, feelings, money spent). But not many potentially predictive variables were assessed, and the ones that were included in the survey (e.g., impulsivity) were not evaluated using validated measures but ad hoc questions, which undermines their validity as potential predictors.

Nonetheless, we have explored the possibility of performing multivariate linear regression analyses for predicting the number of risk factors for pathological gambling, and multivariate logistic regression models to explore if some sociodemographic variables could predict screening positive in the BQPG, but none of the models tested was statistically significant. The only variable that had some predictive power was ‘Gender’, as has been previously reported by many different studies.

We have added the analysis description in lines 142-145:

Multiple linear and logistic regressions were calculated to explore the predictive role of the studying sports sciences along with sociodemographic variables on the number of risk factors for pathological gambling, and on screening positive in the BQPG.

The results of such analyses are now available in lines 294-298:

The regression models showed that studying sports sciences was not a significant predictor of the number of risk factors presented for pathological gambling (BQPG + 4 additional items) nor of screening positive in the BQPG. The only sociodemographic variable which consistently predicted both outcomes was gender (B = 0.64, p < .001 for risk factors and B = 2.90, p = .006 for the screening).

In the discussion, a brief mention to this result has been included (lines 363-367):

In this regard, while studying sports sciences and other sociodemographic variables were not significant predictors of the number of risk factors nor of screening positive in pathological gambling, gender did play a significant role in both cases, confirming the higher prevalence of this problem among men, even in the university students population [34], which could be the reason behind the differences among the two sub-groups that we compared.

And we have added that our study design presents limitations regarding the variables included and the analyses that were performed (lines 381-383):

Future studies could include those variables and analyze how they can predict the frequency of usage of different gambling options, the money spent, or the risk of developing pathological gambling.

- Could the authors discuss the possible reasons that the current sample has more betting online use than the prevalence of betting online in the general population in Spain in the last 12 months in 2019 (both, in the 15-65 years old group and 14-18 years old group; according to the "Observatorio Español de las Drogas y las Adicciones, 2020")

Authors: Thank you for raising this up. We believe that these are not fully comparable results since our survey did not ask about the previous 12 months but about the lifetime (“have you ever…?”) and this partially justifies a higher prevalence of any behavior, compared to the one that would have been reported if the participants had been asked regarding a particular time frame.

In addition, our study is focused on young individuals (mean age 20.57) while the 2020 report includes much wider age ranges in the first case (15-65) and only teenagers (14-18), who are legally not allowed to gamble, in the second case. Our age group is legally free to gamble, so it is expectable that their prevalence of betting online is higher compared to teenagers. Also, they are very familiar with new technologies which could enhance their interest in betting apps and online platforms and, as explained in the introduction, they constitute a population in risk of doing so because of their financial situation which can make them see in gambling an opportunity for earning easy money. These two factors could justify that university students could use online betting more frequently than the general population (aged 15-65).

We have included a new paragraph reflecting the main ideas in lines 301-311:

According to our results, betting shops (44.2%) was the most common option used for gambling, much more than online betting (11.2%). This goes in line with the 2020 report on behavioral addictions published by the Spanish Ministry of Health [1], which observed that in-person gambling options such as betting shops had been much more commonly used (63.6%) than online betting platforms (6.7%) in the previous 12 months. While these results are not fully comparable to the ones of the present study due to the different age groups that are analyzed (i.e., university students vs. people aged 15-64) and the time frame referred (i.e., lifetime vs. previous 12 months), they seem to suggest that online betting is still not as popular as the traditional in-person gambling options, and therefore the preventive strategies should not forget these targets despite the increasing significance of online platforms in behavioral addictions.

- The power to generalize the current results to the other populations may be affected as only university students were included (and was a convenience sample). Hence, the authors should state this as a limitation, as the current results may be only interpreted in specific similar samples.

Authors: We completely agree with the reviewer, this has been included in lines 368-370:

First, the study used a convenience sample including students of Aragon, Spain, and therefore the generalizability of the results is limited.

Reviewer 2 Report

This paper presents the findings of a survey of 516 Spanish university students to gain insights into the prevalence of gambling; problem gambling; and, the potential influence of advertising on young people.

The study shows that around half of the respondents had gambled in their life-time and that sports betting was popular amongst some participants. Some reported receiving or being exposed to gambling advertising. Some comparisons are provided concerning the difference between students from sports science and other disciplines. The paper presents a lot of descriptive statistics in a small number of tables, but mostly in the text.

I had quite a few difficulties with this paper.

First, it needs some editing. An experienced English language specialist should proof-read the paper to correct some of the writing, e.g., ‘youngsters’ (slang); ‘gambling advertisings’; ‘being superior to E30’, etc..

Second, the paper does not really add that much to the literature. Many studies of this nature have been undertaken and this is really only a convenience sample in one location. The sample is not large enough to stand out from many other studies. More needs to be said about how representative Uni students are compared with the general population. In other countries, students often spend less on gambling. In the US, it may be higher due to the interest in sports at some colleges.

Third, the measure of problem gambling would not generally be considered valid in the international literature.

Fourth, the study has far to much descriptive information and does not control for gender in the comparisons between sports science and other disciplines. Sports science has more males and males are more likely to gamble? There needs to be some modelling that controls this. Not enough information is provided is provided on the breakdown of frequency of gambling participation and the exposure to advertising. The information seems fragmented and hard to follow.

Fifth, the paper often slips into a form of moral panic which a focus on the negative aspects of gambling. In fact, very little is being spent per week. Gambling IS healthy and normative if participation rates are within safe limits- I don’t see what it necessarily has to be made to be a dangerous behaviour for all people. Modern countries need to have regulated legalized gambling.

Sixth, microtransactions are not clearly defined. How do these feature in gambling? Exotic bets?

Author Response

First, it needs some editing. An experienced English language specialist should proof-read the paper to correct some of the writing, e.g., ‘youngsters’ (slang); ‘gambling advertisings’; ‘being superior to E30’, etc..

Authors: Thank you for pointing this out. Following the reviewer’s suggestion, the manuscript has been edited by a native English speaker.

Second, the paper does not really add that much to the literature. Many studies of this nature have been undertaken and this is really only a convenience sample in one location. The sample is not large enough to stand out from many other studies. More needs to be said about how representative Uni students are compared with the general population. In other countries, students often spend less on gambling. In the US, it may be higher due to the interest in sports at some colleges.

Authors: This study was conducted on a population (university students from Aragon) for which the guidance and tutoring services had reported an increase in cases of pathological gambling. The Spanish Ministry of Health, in the 2020 report, pointed out that gambling was becoming a major concern; this report indicated that approximately 2 out of 3 Spaniards (aged 15-64) had gambled in the previous 12 months. This has been added in the Introduction, lines 39-40:

According to the Spanish government, in 2020, 64.2% of the Spanish population ad-mitted having gambled at least once in the previous 12 months [1].

In the Discussion, we have compared the results of our study with the ones reported for the general population, highlighting the differences between both samples (lines 301-311):

According to our results, betting shops (44.2%) was the most common option used for gambling, much more than online betting (11.2%). This goes in line with the 2020 report on behavioral addictions published by the Spanish Ministry of Health [1], which observed that in-person gambling options such as betting shops had been much more commonly used (63.6%) than online betting platforms (6.7%) in the previous 12 months. While these results are not fully comparable to the ones of the present study due to the different age groups that are analyzed (i.e., university students vs. people aged 15-64) and the time frame referred (i.e., lifetime vs. previous 12 months), they seem to suggest that online betting is still not as popular as the traditional in-person gambling options, and therefore the preventive strategies should not forget these targets despite the increasing significance of online platforms in behavioral addictions.

Moreover, as explained in the introduction, university students constitute a particularly relevant subgroup in this regard, mainly because of their age: they are more likely to see gambling as a fun, normal leisure activity, and even consider it a potentially profitable activity. We believe this justifies that we focused our research on this population, although we completely agree that this was a convenience sample and that this implies some limitations (this has been stated in lines 368-369):

First, the study used a convenience sample including students of Aragon, Spain, and therefore the generalizability of the results is limited.

Third, the measure of problem gambling would not generally be considered valid in the international literature.

Authors: We agree with the reviewer, and this has been added as a limitation (lines 373-376):

In this line, it also needs to be noted that the measure used for assessing potential pathological gambling in this study (BQPG) is a screening measure and that a proper diagnostic following the DSM-5 criteria should have been included to increase the validity of the outcome.

Fourth, the study has far too much descriptive information and does not control for gender in the comparisons between sports science and other disciplines. Sports science has more males and males are more likely to gamble? There needs to be some modelling that controls this. Not enough information is provided on the breakdown of frequency of gambling participation and the exposure to advertising. The information seems fragmented and hard to follow.

Authors: We must agree with the reviewer, the study contains a lot of descriptive information, but this is because that was its original aim. We have rewritten this section and included a new table with detailed information about the exposure to betting advertising in an effort to make it easier to read (line 167). We have also performed sensitivity analyses after including as covariates those variables that showed significant differences between the two subgroups. However, no changes in terms of statistical significance were observed after these analyses.

Fifth, the paper often slips into a form of moral panic with a focus on the negative aspects of gambling. In fact, very little is being spent per week. Gambling IS healthy and normative if participation rates are within safe limits- I don’t see what it necessarily has to be made to be a dangerous behaviour for all people. Modern countries need to have regulated legalized gambling.

Authors: We understand the points raised by the reviewer, and while we agree that gambling may be a normative activity and is not problematic for many individuals, we believe that the results of our study (and previous ones) are clear when indicating that there is a significant number of university students who are struggling with pathological gambling. Addictive behaviors such as this have been declared a “public health problem” by the Spanish government. The money spent might be little at first, although many university students do not have any income and therefore, proportionally those quantities would not be so little. Moreover, the development of an addiction would eventually increase these quantities of money. Gambling is already perceived as normal by a very notable part of our society (in our opinion, because of the efforts of betting marketing), and thus the tone of our writing was voluntarily addressed at highlighting the risks of gambling and the need to regulate it and to design strategies to prevent the development of addictive behaviors.

Sixth, microtransactions are not clearly defined. How do these features in gambling? Exotic bets?

Authors: We agree with the reviewer that some definition on this regard was needed. It has been included in lines 69-73:

Along with the classic gambling options (e.g., betting shops, casinos, bingos), in the last decade some other types of gambling, mainly delivered via Internet, have become increasingly popular, mainly among young people. That is the case of micro-transactions, commonly included in online videogames, which consist of small payments, sometimes in the form of a bet, to obtain a virtual good.

Round 2

Reviewer 1 Report

The authors performed an incredible work, the manuscript has been improved and gained in quality. All the suggestions and comments were addressed. 

Author Response

Thank you very much for your comments and suggestions.

The manuscript has now been revised by a native English speaker.

Reviewer 2 Report

This version has all the same flaws as the original version.

Author Response

Following the reviewer's suggestion, the manuscript has been deeply revised by a different Englihs native speaker.   We tried to address all the comments that the reviewer shared, which implied including numerous modifications in the manuscript: the comparison of our sample with the general population, the limitation of using a screening measure to measuring problem gambling, a new table with information about the exposure to advertising, and a definition of micro-transactions.
